# Effect of Prepartum Magnesium Butyrate Supplementation on Rumination Time, Milk Yield and Composition, and Blood Parameters in Dairy Cows

**DOI:** 10.3390/vetsci10040276

**Published:** 2023-04-05

**Authors:** Hedvig Fébel, Joan Edwards, Ferenc Pajor, Viktor Jurkovich, Mikolt Bakony, Levente Kovács

**Affiliations:** 1Institute of Physiology and Nutrition, Kaposvár Campus, Hungarian University of Agriculture and Life Sciences, Gesztenyés út 1, H-2053 Herceghalom, Hungary; 2Palital Feed Additives B.V., De Tweede Geerden 11, 5334 LH Velddriel, The Netherlands; 3Institute of Animal Sciences, Gödöllő Campus, Hungarian University of Agriculture and Life Sciences, Páter Károly utca 1, H-2100 Gödöllő, Hungary; 4Department of Animal Hygiene, Herd Health and Mobile Clinic, University of Veterinary Medicine, István utca 2, H-1078 Budapest, Hungary; 5Department of Biostatistics, University of Veterinary Medicine, István utca 2, H-1078 Budapest, Hungary; 6Bona Adventure Ltd., Peres utca 44, H-2100 Gödöllő, Hungary

**Keywords:** magnesium butyrate, dairy cows, rumination time, haptoglobin, milk composition, transition period, blood energy analytes

## Abstract

**Simple Summary:**

Prepartum magnesium butyrate (MgB) supplementation was given to dairy cows three weeks before parturition and lactational performance; inflammation-related protein levels, blood energy analytes, and rumination time were compared to Control cows. Milk samples were collected for the first 70 days of lactation, and various blood parameters and rumination activity were also monitored. The MgB-supplemented group yielded more milk compared to the Control during week 1, and had increased milk fat and protein levels over a longer duration of lactation. Blood energy analytes indicated that MgB supplementation had no effects on the negative energy balance that occurs often during the early lactation phase. Whilst no clinical disease was detected in either group of animals during the study, the MgB group was found to have a significantly lower milk somatic cell count and blood concentration of an inflammation-related protein. Rumination activity was greater in the MgB group relative to the Control group due to a depression in rumination activity being overcome. The basis of this is not clear, as feed intake could not be monitored in the current study.

**Abstract:**

Background: Magnesium butyrate (MgB) supplementation of dairy cows during the three-week close-up period was tested for its effects on blood energy analytes, rumination time, inflammation, and lactation performance. Methods: Daily milk yield was recorded and weekly milk samples collected for the first 70 days of lactation from MgB supplemented (MgB, n = 34), and unsupplemented (Control, n = 31) multiparous Holstein-Friesian cows. During a period from week 3 to week 10 postpartum, blood samples were taken and analyzed for various parameters, and ruminant activity was measured. Results: The MgB group yielded 25.2% more milk than the Control during week 1, and had increased milk fat and protein concentrations over a longer duration. Somatic cell counts (SCC) were decreased in the MgB group independent of days in milk. No differences were observed between groups in terms of plasma non-esterified fatty acids, β-hydroxybutyrate, glucose, or blood iCa levels. The MgB group had lower haptoglobin (Hp) levels during lactation relative to the Control group. Time spent ruminating increased after calving with MgB due to a shorter post calving rumination delay relative to the Control group. Conclusions: Prepartum MgB supplementation improved lactation performance without affecting blood energy analytes. The basis by which MgB also improved rumination activity remains to be determined, as DMI could not be assessed. As MgB lowered SCC and Hp concentrations, it is speculated that MgB may help minimize postpartum inflammatory processes.

## 1. Introduction

In the last 40 years, several major scientific advances have been realized in terms of determining the opportunities and limitations of changing milk chemical composition using nutritional manipulation [1,2,3]. The type of ration fed to cows often affects volatile fatty acid (VFA) concentrations in the rumen, which may in turn affect milk yield and composition. The development of the rumen wall is stimulated by VFAs (i.e., acetate, propionate, and butyrate) which increase cell proliferation in the rumen wall tissue [4]. During the dry period, lower VFA production, from rations typically fed to dairy cows during this time, is associated with a decreased rumen wall surface area relative to during lactation [5]. The length of time taken for full redevelopment of rumen papillae following calving has been suggested to be influenced by diet [6]. As an inhibitor of cell apoptosis in rumen wall tissue, butyrate induces a significant improvement in papillae length relative to other ruminal VFAs [7], and can also increase VFA transportation across the rumen epithelium [8,9]. As such, butyrate stimulates rumen redevelopment via multiple mechanisms.

Butyrate in the rumen is mainly metabolized by the epithelium, forming ketone bodies and CO_2_ [10]. Ketone bodies are also associated with intensified mobilization of body fat reserves, with circulating non-esterified fatty acids (NEFA) and β-hydroxybutyrate (BHB) typically being elevated during early lactation. This is an adaptive response to negative energy balance (NEB) in order to supply peripheral tissues with alternative energy sources and spare glucose for milk synthesis [11]. Over the last 25 years, NEFA and BHB have become the accepted biomarkers of excessive NEB or maladaptation of the energy metabolism to lactation. As such, antepartum and postpartum thresholds have been established to predict the risk of NEB-related diseases and associated milk loss in transition dairy cows [12]. As well as this challenge, during the transition period immune competence is also reduced, and systemic inflammation commonly occurs which negatively impacts cow performance [13].

When the immune system is challenged, blood concentrations of certain acute phase proteins (APPs) change [14]. Haptoglobin (Hp) is an APP that is widely accepted as a biomarker for systemic inflammation, and is released into the bloodstream in response to tissue damage, inflammation, infection, and bacterial components, as well as stress [15,16,17,18]. Hp is normally very low or absent in healthy cows [19]; however, cell-level acute-phase responses can occur during the stressful transition phase. As butyrate can beneficially modulate inflammation via multiple mechanisms [20], butyrate supplementation of transition cows is of interest. However, as previously discussed by Engelking et al. [21], studies to date have been inconclusive and it may be that the effect of butyrate is influenced by diet.

In this study, the effect of prepartum magnesium butyrate (MgB) supplementation on periparturient dairy cows was investigated, as the effect of this form of butyrate on inflammation and lactation of dairy cows is not known. Measurements taken during the study prepartum and/or during the first 70 days in milk (DIM) included milk yield and composition, blood energy analytes, blood ionized Ca (iCa), Hp concentrations, and rumination time.

## 2. Materials and Methods

### 2.1. Study Design, Animals, Sampling, and Ethical Approval

All methods and the procedures applied on the animals were performed in accordance with the relevant guidelines and regulations of the Pest County Government Office, Department of Animal Health (Permit Number: PE/EA/1973-6/2016) that approved the study.

The study was carried out on a large-scale dairy farm in Hungary (47°18’191’’ N 18°48’336’’ E) during a period from March to September 2021. Eighty multiparous Holstein-Friesian cows were blocked by their expected calving date (between April and June) and randomly assigned to one of two parallel groups: Control or MgB. The cows were all clinically healthy at the start of the trial, had a body condition score (BCS) between 3 and 4.2, and a locomotion score of 2 or lower [22].

From three weeks prior to expected calving, cows in the MgB group (n = 40) received prepartum diets containing MgB supplementation (MgB). The cows in the Control group (n = 40) received the same prepartum diet as the MgB group but without supplemental MgB. No treatment was given after calving, i.e., lactation rations for both groups were identical. Detailed dietary-related information is provided in Section 2.2 and Section 2.3. During the study, milk and blood samples were collected and analyzed as outlined in Section 2.4, Section 2.5, Section 2.6 and Section 2.7. Daily milk yield was recorded during the first 70 DIM using on-farm records from the milking systems. Rumination activity during the trial was monitored using the RuminAct^®^ acoustic biotelemetry system (SCR Engineers Ltd., Netanya, Israel) as outlined in Section 2.8.

To reach the appropriate number of prepartum blood samples per animal, and the intended duration of prepartum MgB supplementation in the case of MgB cows, animals with less than 18 days or more than 25 days in the prepartum part of the trial were excluded from the study. This meant that in the final dataset only 31 Control (means ± SD; parity = 2.96 ± 0.13 (parity 2; n = 13, parity 3≤; n = 18), previous lactational milk yield = 9705 ± 148 kg, BCS = 3.56 ± 0.02) and 34 MgB cows (parity = 2.98 ± 0.12 (parity 2; n = 16, party 3≤; n = 18), previous lactational milk yield = 9762 ± 163 kg, BCS = 3.48 ± 0.02) were used in the experiment. The monthly average temperature-humidity index (THI) and the number of animals starting monthly in each trial group is presented in Table 1.

### 2.2. Animal Management, Housing and Diets

From 28 days before expected calving, MgB and Control cows were housed in their respective group pens with a stocking density of 12 m^2^/animal. Pens were bedded with deep straw. Before calving, cows were fed a prepartum total mixed ration (TMR) ad libitum containing a dietary forage to concentrate ratio of 81.7:18.3 on a dry matter (DM) basis (Table 2). The prepartum TMR was fed twice daily (early morning and late evening). Three weeks before the expected time of calving, 70% MgB encapsulated in a fat matrix (Rumen-Ready^®^, Palital Feed Additives B.V., Velddriel, The Netherlands) was added to the pre-calving TMR of MgB cows before morning feeding according to the number of cows in the MgB group. A dosage level of 150 g/cow/day (i.e., included at 1.15% DM) was used, which is equivalent to 105 g MgB/cow/day. This dosage was selected based on preliminary trials which showed favorable results at this feeding level (data not shown). Rumen-Ready^®^ was added directly into the TMR during its preparation.

Calvings took place in the prepartum group pen or, if continuous supervision or obstetrical assistance was required at calving, in a separate maternity pen. During the first 5 DIM, cows were housed in postpartum pens (each including 4 animals) and were milked twice daily at 0400 and 1400 h in a 4-stall herringbone milking parlor operated with DeLaval Control Valve bucket milking machines (DeLaval International AB, Tumba, Sweden). After 5 DIM, cows were introduced to the fresh lactation group and milked twice daily at 0500 and 1500 h in a 2 × 28-stall parallel Bosmark milking parlor (Bosmark Kft, Biatorbágy, Hungary). Cows were fed a postpartum TMR ad libitum, with a 57.7:42.3 forage to concentrate ratio on a DM basis until 90 DIM (Table 2), and water was available ad libitum. The postpartum TMR was delivered twice daily, according to the milking schedule. The cows found the freshly delivered TMR in the barn when arriving back from milking. Prepartum and postpartum diets did not change during the study period and were both fed as a total mixed ration (i.e., no additional concentrates were fed via the milking systems). Ingredients and chemical composition of both the prepartum and postpartum TMR diets are shown in Table 2. Dry matter intake (DMI) of cows housed in the prepartum pens was estimated daily before early morning feeding for control and MgB groups by subtracting the orts from the amount of feed offered, and dividing by the number of cows per pen. However, this approach to estimate DMI was not possible postpartum as study animals from Control and MgB groups were no longer housed separately.

### 2.3. Diet Analysis

Chemical composition of the prepartum and postpartum TMR diets was analyzed with a 4-week sampling frequency from the start of MgB feeding until 70 DIM of the last cow enrolled to the study. On the day of sampling, for each TMR, four samples (i.e., two sub samples of the TMR fed in the morning, and two sub samples of the TMR fed in the evening) for laboratory analysis were collected as follows. Immediately following TMR distribution, a bucket (10 L) was filled with handfuls of TMR from the top, middle, and bottom of the TMR from the entire length of the feeding path. The bucket contents were thoroughly mixed, and two sub samples made. The TMR samples were then dried after collection. For analysis, parameters were selected in order to confirm that the differences in nutrient composition between the prepartum and postpartum diets were as expected.

Immediately before laboratory analysis, feed samples were dried at 60 °C for 48 h until constant weight and ground through a 1 mm screen using a sample mill (Cemotec, Tecator, Sweden). The ground samples were analyzed for the various contents as follows. N content was analyzed according to the Association of Official Analytical Chemists (AOAC) [23] method 984.13. Ash was analyzed using AOAC method 942.05 [23]. Acid detergent fiber (ADF) content was analyzed using AOAC method 973.18 [23]. Crude fiber content was analyzed using AOAC method 962.09 [23]. Ether extract was analyzed using AOAC method 920.39 [23]. Neutral detergent fiber (NDF) content was analyzed using α-amylase and sodium sulfite as previously described by Van Soest et al. [24]. Acid detergent lignin (ADL) was also analyzed according to the method described by Van Soest et al. [24]. Starch content was analyzed by an enzymatic colorimetric method described by Bach Knudsen et al. [25]. Sugar was measured by the enzymatic colorimetric method described by Larsson and Bengtsson [26]. The acceptable recovery of the methods is a function of the concentration. At the concentration of 100%, 10%, 1% and 0.1% acceptable recovery requirements for the individual assays listed above are 98–101%, 95–102%, 92–105%, and 90–108%, respectively.

### 2.4. Milk Sampling and Composition Analysis

For each cow, milk samples were collected during the morning milking weekly after calving from all four udder quarters and pooled. The first morning milk sample was collected on day 1 after calving, and was a minimum of 12 h after calving. The collected milk was then transferred into plastic graded milk buckets and was agitated with a whisk/mixing spoon for at least 2 min, to ensure an even distribution of constituents, before a 100 mL sample was collected and divided into two large plastic vials (50 mL each). One 50 mL vial was used for fat, protein, lactose, and total solids contents analysis, and the second 50 mL vial for somatic cell count (SCC) measurements. All samples were frozen at –20 °C until analysis. Frozen milk samples (50 mL) were thawed in a water bath at 40 ± 2 °C, and vials were inverted 10 times to thoroughly mix the milk and give an even distribution of constituents. Milk fat, protein, lactose, and total solids were determined using the LactoScope™ infrared spectrometry analyzer (Delta Instruments, Drachten, The Netherlands). The SCC was determined using the Bentley FCM apparatus (Bentley Instruments Inc., Chaska, MN, USA).

### 2.5. Plasma BHB, NEFA, and Glucose

Blood samplings were performed weekly: 4 samples before calving (from day 255–260 of pregnancy until calving, i.e., day–21, day–14, day–7 and day–1), one sample right after calving (day 0) and 5 samples (i.e., 7, 14, 21, 35 and 70 days after calving) covering the first 70 DIM. Blood was taken at approximately 0900 h during the prepartum period, and 1 h after morning milking during the postpartum period. In both periods, this relates to a sampling time approximately 1 h after the morning feeding. Blood samples were taken by puncture of the median coccygeal vein and collection into vacuum tubes (Vacuette, Greiner Bio-One, Kremsmünster, Austria) containing either sodium fluoride (for glucose analysis) or K-EDTA (for NEFA and BHB analysis).

Samples were cooled to 4 °C and then centrifuged (10 min, 3000× *g*) within 60 min of collection. Plasma was then harvested immediately and stored at –75 °C until analysis. The plasma glucose, NEFA and BHB were analyzed with commercial kits. Enzymatic, colorimetric tests were used to obtain plasma glucose (kit no. 46862, Diagnosticum Ltd., Budapest, Hungary) and NEFA concentrations (kit no. FA 115, Randox Laboratories Ltd., Crumlin, UK), whereas plasma BHB levels were measured using an enzymatic, kinetic UV test (kit no. RB 1008, Randox Laboratories Ltd., Crumlin, UK).

### 2.6. Serum Haptoglobin

Directly following blood collection for metabolic parameter analysis, blood was collected for haptoglobin analysis from the median coccygeal vein into vacuum serum tubes (Vacuette, Greiner Bio-One, Kremsmünster, Austria). Blood was allowed to clot for up to 2 h at room temperature, and then was centrifuged at 1000 rpm for 10 min at room temperature for serum collection. The collected serum was stored in a cryovial at −75 °C until it was assayed for Hp. Concentrations of Hp were determined using a commercially available bovine Hp test kit (P801; Tridelta Development Ltd., Maynooth, Ireland) following the manufacturers guidelines. The Hp test kit uses a solid phase ELISA assay with affinity purified anti-bovine Hp antibodies for solid phase immobilization, and horseradish peroxidase conjugated anti-bovine Hp antibodies for detection.

### 2.7. Ionized Calcium

At the time of blood samplings for blood energy metabolites and Hp measurement, blood was also collected with syringes containing a Ca^2+^ balanced lithium heparin preparation (Blood Gas Monovette^®^ 2 mL LH, Sarstedt, Nümbrecht-Rommelsdorf, Germany). Within 5 min of collection, blood was tested using the ABL800 Basic system (Radiometer Medical ApS, Brønshøj, Denmark) for iCa concentrations (i.e., Ca^2+^; mmol/L).

### 2.8. Rumination Time

Rumination activity during the trial was monitored using the RuminAct^®^ acoustic biotelemetry system (SCR Engineers Ltd., Netanya, Israel). This system is validated for the measurement of the rumination activity of dairy cows [27], and has also previously been used for dairy cows around parturition [28,29,30]. The system consists of rumination sensors, stationary readers, and software for processing the electronic data, which are subsequently transferred to a computer using a wireless connection. The RuminAct^®^ system sensor contains a microphone for recording rumination sounds, and a microprocessor for data conversion.

RuminAct^®^ neck collars were fitted to the cows during their movement to the prepartum pen, which was 28 days before the expected time of calving, and the system was used to continuously record rumination activity until 70 DIM. According to the manufacturer’s recommendations, RuminAct^®^ sensors needed 2–3 days to become familiarized with the animals’ individual rumination pattern. For each animal, the raw data set was used to evaluate the length of single rumination periods and breaks between rumination periods. This enabled the start and end of rumination periods to be labeled with a time resolution of 1 min, and then rumination time (min/d) was calculated.

### 2.9. Statistical Analysis

The R statistical software was used for all data visualization, estimations, and hypothesis testing [31]. Data were tested for constant variance (Levene’s test) and the Shapiro–Wilk test was used for testing the equality of error variances. The level of significance was in all tests set at *p* < 0.05.

Milk composition and milk yield were compared weekly between Control and MgB groups during the first 70 DIM. Daily milk yield measurements were averaged over weeks, whereas parameters of milk composition were determined once a week. Mean milk yield, lactose, fat, protein, and total solids percent and SCC were compared by fitting a linear mixed model (random intercept model) with each of the parameters as a response variable. Treatment, time (i.e., week), and their interaction term were fixed factors in the model. The SCC data were transformed into logarithmic forms before performing statistical procedures. Cow ID was added as a random term to account for repeated measures on the same animal. Groups were compared at each week of sampling using contrasts. Single-step multiple comparisons were based on studentized range distributions, controlling for the family-wise error rate using Tukey’s *p*-value adjustment.

Blood parameters (i.e., BHB, NEFA, glucose, Hp, and iCa concentrations) were also compared between Control and MgB groups regarding each sampling occasion (−21, −14, −7, −1, 0, 7, 14, 21, 35, and 70 days postpartum). In case of non-homogenous variance of residuals and non-normality of the random effects, log transformation was used to achieve conditions of applicability of the linear mixed model. Treatment, time (i.e., day of sampling), and their interaction term were the fixed effects, with cow ID included as a random effect to account for repeated measures on the same animal. Separate models were fitted to each of the investigated parameters as response variables. Groups were compared controlling for day of sampling by post-hoc comparison of contrasts with Tukey’s adjustment. The ‘lme4′ [32], ‘emmeans’ [33], and ‘lawstat’ [34] packages of the R statistical environment were used for fitting linear mixed models and multiple comparisons.

Areas under the daily rumination time curves (AUC) were calculated for prepartum (between 23 days before calving and birth) and postpartum periods (between calving and 70 DIM) for each individual, and the averaged values were used for comparisons across groups. To determine AUC_RUM_, a trapezoid method was used [35] as follows:AUC = Σ [(R_n_ + R_n+1_)/2 × h—BASELINE],
where ‘R’ is a value of the time spent ruminating at a given day, ‘h’ is the time in days between the two R-values, and ‘baseline’ is the mean value of the time spent ruminating, calculated for the first four days of the measurement. Baseline values, AUCs, and time to return to baseline were compared between groups with the Welch’s two-sample *t* test.

## 3. Results

The cows in the MgB group spent 23.6 ± 0.8 days (range; 18–25 days) in the prepartum pen in treatment, whereas the cows in the Control group spent 23.9 ± 1.0 day (range; 21–25 days) in the prepartum pen before calving.

### 3.1. Milk Yield and Milk Chemical Composition

Changes in weekly averages of daily milk yield of the Control and MgB groups and milk chemical composition for milk samples collected on weeks 1, 2, 3, 4, 5, 6, 7, 8, 9, and 10 are presented in Figure 1a–f. The milk yield increased from calving in both groups with an effect of time (*p* < 0.001), and reached its maximum volume on week 4 in both groups (Figure 1a). The overall treatment effect was not significant (*p* = 0.439), but there was a treatment × time interaction as milk yield was 25.2% more in the MgB group compared to the Control group on week 1 (*p* < 0.001).

Milk fat concentration decreased from week 1 to 10 (time effect: *p* < 0.001) with a similar pattern in both the MgB and Control groups (Figure 1b). There was a significant treatment effect (*p* < 0.001) and a significant treatment × time interaction (*p* < 0.001), indicating greater fat concentration in the milk of the MgB group sampled on weeks 1, 2, 3, 8, and 9, relative to the Control group. Similar to milk fat, milk protein levels also decreased from week 1 to 10 (*p* < 0.001) in both groups (Figure 1c). A significant treatment effect (*p* < 0.001) and treatment × time interaction (*p* < 0.001) occurred, with increased milk protein concentration for the MgB group compared to the Control group on weeks 1, 2, 3, and 10. Lactose concentrations increased within the first 6 weeks after calving, then stabilized for a few weeks before decreasing again in both groups (Figure 1d). Whilst this time effect was significant (*p* < 0.001), treatment had no effect on lactose concentration (*p* = 0.389) and there was also no treatment × time interaction (*p* = 0.393). The total solids percent decreased after week 1 in both groups (Figure 1e) and there was a time effect (*p* < 0.001), an overall treatment effect (*p* < 0.001), and a treatment × time interaction (*p* < 0.001). Total solids percent was greater in the MgB group in milk sampled on weeks 1, 2, 3, 5, 8, 9, and 10 compared to milk from the Control group (Figure 1e). Time of sampling had a significant effect on SCC (*p* = 0.851) and there was a significant treatment effect (*p* = 0.022). The treatment × time interaction (*p* = 0.406) was not significant, and multiple comparisons showed that SCC on weeks 1 and 3 were significantly lower in milk samples from the MgB group compared to the Control group (Figure 1f).

### 3.2. Blood Analytes

Group means of all blood analytes according to day of sampling and respective *p*-values of treatment, time, and treatment × time interaction effect are summarized in Table 3. There was a significant time effect on all the metabolic parameters in both groups; BHB levels increased on day 7, and NEFA and glucose concentrations increased on day 0. No difference was observed in plasma BHB, NEFA, or glucose concentrations between the MgB and Control groups, and also no treatment × time interaction occurred. Time and treatment both affected Hp concentrations, and there was a treatment × time interaction. Serum Hp levels were similar between groups until day 0, then increased from day 7 and remained higher throughout the 70 DIM compared to the prepartum period in both groups. During the postpartum period, concentrations of haptoglobin were significantly lower in the MgB group relative to the Control group, which resulted in a treatment × time interaction. There was a significant time effect on blood iCa levels. With the onset of lactation (day 0), blood iCa levels decreased in both groups with an average of 0.1 mmol/L, but stayed within the normal range throughout the measurement period. Ionized calcium levels in the blood did not differ between Control and MgB groups during the measurement period, and no treatment × time interaction occurred.

### 3.3. Rumination Time

Rumination sensors were attached to the Control and MgB cows from 27.1 ± 2.9 days (ranging between 30.0 and 24.1 days) and 27.8 ± 2.7 days (ranging between 30.2 and 24.6 days) respectively before calving until 70 DIM. Changes in rumination time (means ± SEM) are shown in Figure 2 for the Control and MgB groups.

The daily time spent ruminating showed a similar pattern between groups during the prepartum and postpartum periods, except for a delay postpartum in resuming rumination in the Control group relative to the MgB group (starting from day 4 of lactation). There was no significant difference between groups in basal rumination times calculated for the first 4 days of the measurement, and also not in prepartum AUC rumination time (Table 4). However, AUC rumination time was significantly greater in the MgB group than the Control group when calculated for the first 70 DIM (Table 4).

## 4. Discussion

During the present study, cows that received prepartum MgB supplementation (70% MgB product, fed at 1.15% of dietary DM) produced 25.2% more milk during the first 7 DIM compared to the Control group. The improved milk yield might be explained by the known beneficial effects of butyrate on ruminal VFA transport proteins, tissue development, and epithelial blood flow [7,8,9]. However, the effects on DMI cannot be ruled out as DMI was unable to be measured in the current study. Whilst milk yield was only improved during the first week of lactation, longer lasting improvements in terms of the fat and protein content of the milk were found during the monitored 70 DIM. The milk of cows receiving prepartum MgB supplementation was richer in fat and protein and had a higher concentration of total solids. This difference was significant at the first milk sampling, which may have been influenced by the transition from colostrum to milk [36]. However, as the improvement was also significant during the following two weeks of lactation, it seems that the changes in milk composition during the first week are likely to be genuine.

Consistent with the present study findings, butyrate supplementation of the diet of dairy cows (70% sodium butyrate product, fed at 1.1% of dietary DM) improved the 4% fat-corrected milk yield, and markedly increased milk fat [37]. However, it should be noted that in the study of Izumi et al. [37], the butyrate was supplemented during lactation, unlike in the present study where it was only fed prepartum. As such, the underlying mechanism by which the milk composition is altered may well differ between this and the present study. This is due to butyrate being able to be used for de novo synthesis of fatty acids in the mammary glands when it is fed during lactation. Authors also indicated that butyrate supplementation can beneficially influence N metabolism in ruminants as postpartum butyrate supplementation decreased serum urea N and milk urea N. Unlike the present study, though, where prepartum supplementation was used, postpartum butyrate supplementation did not increase milk protein yield. As butyrate stimulates rumen tissue development, Izumi et al. [37] suggested that their findings could be potentially explained by increased rumen tissue synthesis enhancing urea N recycling.

Prepartum sodium butyrate supplementation (30% sodium butyrate product, fed at 1.2% of dietary DM) in the close-up period has been previously reported to have no effect on milk yield [38]. In a recent study, Engelking et al. [21] also failed to find a significant benefit of calcium butyrate supplementation (70% calcium butyrate product, fed at 1.4% of dietary DM) on milk yield or composition when it was supplemented to dairy cows from the close-up period until 24 DIM. The reason for these contrasting findings relative to the present study is not clear. Differences in diets used in the studies may be an influencing factor. The length of time the butyrate was fed, dosage used, or differences in the type of butyrate product used may also play a role. For example, feeding cows additional calcium or sodium prior to calving is likely to have a different effect relative to feeding additional readily available magnesium [39]. The design of this study, however, did not enable the effects of magnesium and butyric acid to be separately assessed to confirm if this was the case. However, it should be noted that the prepartum MgB supplementation did not affect the blood iCa levels in this study, with a similar pattern over time also evident in both groups. The blood iCa levels remained in the reference range of 1–1.25 mmol/L [40] for both pre- and postpartum in both groups, which enables adequate rumen motility for promoting rumen fermentation. 

In terms of milk lactose concentration, no difference was found between the Control and MgB groups in this study. This is consistent with the findings of previous studies that also reported no effects of pre- or postpartum butyrate supplementation on milk lactose concentration [21,37]. This supports the previous conclusion of Jenkins and McGuire [1] that changes in milk lactose concentration occurs very seldom when cows are fed diets that are within the normal nutritional range.

In the present study the milk SCC was within a normal range, consistent with all animals being clinically healthy. However, prepartum MgB supplementation significantly decreased the milk SCC. Greater concentrations of pre-calving BHB and glucose were previously reported to be associated with lower SCC at first test-milking [41]. However, in the present study no elevations in BHB or glucose were observed for the Control or MgB groups before (or after) parturition. Previous reports have suggested that dietary supplementation of sodium butyrate during lactation could significantly decrease the mammary gland inflammation of ruminants [42,43]. In addition to this, postpartum sodium butyrate supplementation (1% of dietary DM) decreased the immune cells in blood and the SCC in milk [44]. Wang et al. [45] concluded that the underlying mechanism for this dietary butyrate effect was linked to decreased concentrations of a bacterial peptidoglycan component in rumen fluid and plasma, along with an anti-inflammatory effect that was mediated via decreased histone deacetylase 3 expression [43]. This is not surprising as butyrate is a well-known inhibitor of histone deacetylase activity [45], has immunomodulatory functions [46], and also promotes rumen tissue development which may decrease the risk of bacterial peptidoglycan reaching the blood. However, in the present study only prepartum butyrate supplementation was used; therefore, it is not clear to what extent these underlying mechanisms are directly applicable to the findings reported here. Particularly as haptoglobin levels were only decreased postpartum, and not prepartum when the MgB was physically supplemented.

In healthy cattle, serum haptoglobin is either non-detectable or measurable only in very low concentrations (i.e., below 1.0 mg/mL [19,47,48,49]), unlike in cattle with postpartum health disorders where concentrations are elevated [47,50,51,52]. In the present study, serum haptoglobin concentrations were consistently below 1.0 mg/mL, which is in agreement with all study animals being assessed as being clinically healthy. Blood serum Hp levels peaked at day 7 in both groups in the first week postpartum, which is comparable to the findings previously reported by Cartes et al. [53] for transition cows. As such, the findings of this study suggest that, despite the higher postpartum Hp levels of the Control group compared to the MgB group ([Hp] = 0.66 vs. 0.50 mg/mL), there is no clear evidence of a lower postpartum health risk following prepartum MgB supplementation. However, it is interesting to note that the decrease in blood serum Hp levels associated with MgB supplementation was also consistent with MgB also decreasing milk SCC.

Several studies have linked elevated blood levels of NEFA and BHB to increased incidence of NEB-related diseases in transitioning dairy cows [54,55]. In the present study, the energy balance (in terms of BHB, NEFA, and glucose) was not affected by MgB supplementation, including during the time when the MgB was supplemented. BHB, NEFA, and glucose levels were within the generally accepted prepartum and postpartum physiological ranges in both groups during the study [56]. Plasma NEFA levels slightly increased at −7 and −1 day before calving in both groups, similar to previous studies that demonstrated the plasma NEFA concentrations begin to increase from the last 2 weeks of pregnancy [57] to the last 2 to 3 days [58] before parturition. NEFA and BHB levels peaked at day 0 and 7, respectively, for both groups, and were 50–60% of the critical threshold values determined for cattle [12]. In line with the present study findings, Kowalski et al. [38] also did not find any effects of prepartum sodium butyrate dietary supplementation from 30 days before calving on the prepartum BHB and NEFA levels measured 5, 10, and 15 days before parturition.

In the present study, rumination time showed a similar behavior relative to transition as has been previously reported [29,59]. Rumination time decreased during the last two days before parturition in both groups, and then started to increase again in lactation after a delay. However, a greater delay occurred in the Control group compared to the MgB group (day 4 vs. day 18 after calving). Schirmann et al. [60] reported the start of a slow recovery of rumination activity at 6 h after calving, whereas Kovács et al. [29] observed it at 8 h postpartum. The resumption of rumination activity after calving can vary from 2 [29] to 30 days [61], and is directly linked to the start of feed intake [60,62]. The postpartum rumination time observed for the MgB group in this study was within the range reported in the Literature [63,64]; however, the Control group exhibited a depressed rumination activity as indicated by the significant differences between groups in AUC rumination time postpartum.

A delay in resuming rumination is considered an alert signal and might be related to disturbed feed intake or health disorders [64,65,66]. As health disorders have not been indicated from any of the other measured parameters in the study, this reason for the depressed rumination activity seems unlikely. Therefore, it is speculated that the Control group may have had a decreased feed intake postpartum. Unfortunately, this could not be verified in the current study as it was not possible for DMI to be assessed. If it is true that feed intake was increased in the MgB group, the underlying reason for this remains to be determined.

## 5. Conclusions

Prepartum MgB supplementation of dairy cows had a beneficial effect on milk yield and composition. MgB supplementation had no effect on blood energy analytes during the time it was fed prepartum, or afterwards during lactation. Based on the milk SCC and serum Hp concentration findings, it is possible that supplementation of MgB during the close-up period may minimize the risk of sub-acute inflammatory processes developing during lactation. The impact of MgB on DMI could not be measured in the present study, and this limits the ability to understand the basis by which MgB was observed to overcome a depression in rumination activity postpartum.

## Figures and Tables

**Figure 1 vetsci-10-00276-f001:**
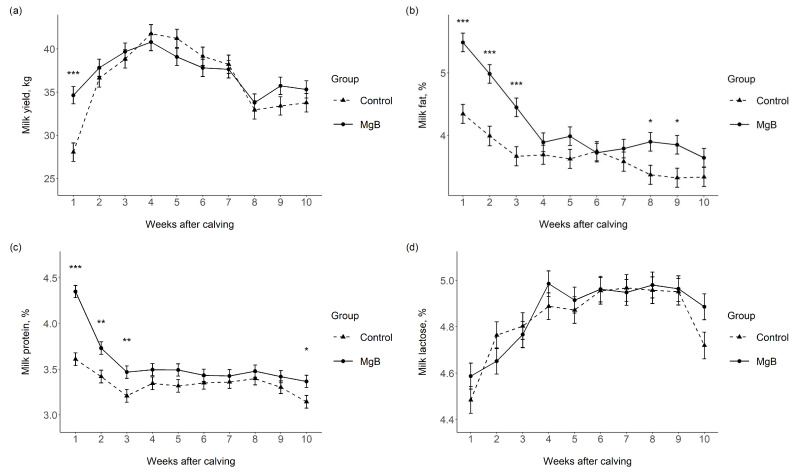
Changes in milk (**a**) yield, (**b**) fat, (**c**) protein, (**d**) lactose, (**e**) total solids, and (**f**) somatic cell count over the first 10 weeks of lactation of dairy cows with (MgB, n = 34) and without (Control, n = 31) prepartum magnesium butyrate supplementation. Data are presented as least squares means and standard errors. For SCC, estimates are back transformed from the logarithmic scale. Significant differences are indicated on individual time points within each chart (* *p* < 0.05; ** *p* < 0.01; *** *p* < 0.001).

**Figure 2 vetsci-10-00276-f002:**
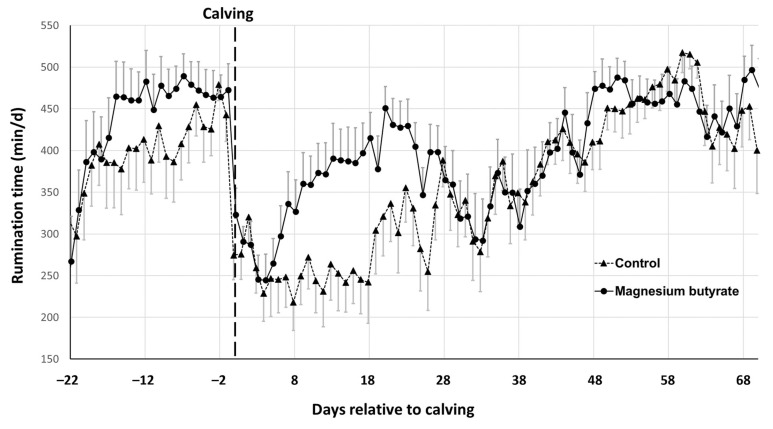
Daily rumination time from 22 days before calving to 70 days in milk of dairy cows with (MgB, n = 34) and without (Control, n = 31) prepartum magnesium butyrate supplementation. Data are presented as daily average means ± SD.

**Table 1 vetsci-10-00276-t001:** Average temperature humidity index (THI) values and the number of animals enrolled in the study during the study period in the barns of magnesium butyrate (MgB; n = 34) and Control (n = 31) groups.

Month	Average THI		No. of Animals Enrolled	
	Control	MgB	Control	MgB
February	42	40	4	5
March	47	47	6	6
April	50	50	7	8
May	58	58	6	7
June	70	71	5	5
July	72	72	3	3

**Table 2 vetsci-10-00276-t002:** Ingredient and chemical composition of prepartum and postpartum total mixed ration (TMR) diets of cows in the Control (n = 31) and magnesium butyrate (MgB) (n = 34) groups.

Item	Prepartum TMR		Postpartum TMR
Ingredient	Control	MgB ^1^	
Forage components (% DM)			
Corn silage	61.8	61.8	60.2
Rye hay	-	-	10.8
Alfalfa baled silage	10.9	10.9	12.9
Grass hay	7.3	7.3	5.3
Straw	12.7	12.7	-
Brewer’s grains	7.3	7.3	10.8
Concentrate components (% DM)			
Corn flour	27.6	27.6	52.0
Oats	13.8	13.8	13.6
Extracted rapeseed meal	13.8	13.8	17.6
Extracted sunflower meal	34.5	34.5	12.0
Premix Mipro Pren 250 ^2^	10.3	10.3	-
Premix Mipro RB 600 ^3^	-	-	4.8
Chemical composition ^4^			
Dry matter (DM), g/kg	437	441	426
Ash, g/kg of DM	84	86	71
Crude protein, g/kg of DM	114	114	167
Ether extract, g/kg of DM	28	29	35
Crude fiber, g/kg of DM	241	232	143
NDF, g/kg of DM	507	494	339
ADF, g/kg of DM	303	280	190
ADL, g/kg of DM	50	47	39
Starch, g/kg of DM	154	158	283
Sugar, g/kg of DM	26	26	29

^1^ Prepartum TMR diet for the MgB group was supplemented with 150 g/cow/day of 70% MgB encapsulated in a fat matrix (Rumen-Ready^®^, Palital Feed Additives B.V., Velddriel, The Netherlands), and was directly mixed into the prepartum TMR during its preparation. ^2^ Contained 18.0% Ca, 2.0% P, 7.5% Na, 8.0% Mg, 5.1% S, 4000 mg/kg Zn, 3000 mg/kg Mn, 900 mg/kg Cu, 93 mg/kg J, 18 mg/kg Se, 12 mg/kg Co, 300,000 IU/kg vitamin A, 100,000 IU/kg vitamin D_3_, 4000 mg/kg vitamin E, 24 mg vitamin B_1_, 12 mg/kg vitamin B_2_, 6 mg/kg vitamin B_6_, 60 μg/kg vitamin B_12_, 360 mg/kg niacin, 24 mg/kg Ca-D-pantothenate, and 12,000 mg/kg choline chloride 15,000 μg/kg biotin (Sano Ltd., Csém, Hungary). ^3^ Contained 16.0% Ca, 0.8% P, 9% Na, 4.2% Mg, 0.4% S, 2500 mg/kg Zn, 2500 mg/kg Mn, 500 mg/kg Cu, 75 mg/kg J, 15 mg/kg Se, 10 mg/kg Co, 200,000 IU/kg vitamin A, 35,000 IU/kg vitamin D_3_, 2000 mg/kg vitamin E, 20 mg vitamin B_1_, 10 mg/kg vitamin B_2_, 5 mg/kg vitamin B_6_, 50 μg/kg vitamin B_12_, 300 mg/kg niacin, 20 mg/kg Ca-D-pantothenate, 48,000 mg/kg choline chloride, 14,000 μg/kg biotin, 0.85% methionine, 6.7% sugar, and 8.4% urea (Sano Ltd., Csém, Hungary). ^4^ Average values of the chemical composition analysis of prepartum and postpartum diets.

**Table 3 vetsci-10-00276-t003:** Blood parameters of dairy cows with (MgB, n = 34) and without (Control, n = 31) prepartum magnesium butyrate supplementation.

		Days Relative to Calving ^1^											*p* Value		
		−21	−14	−7	−1	0	7	14	21	35	70	SEM	Treatment	Time	Treatment × Time
**BHB (mmol/L)**	Control	0.37	0.38	0.36	0.36	0.34	0.71	0.43	0.60	0.60	0.51	0.074	0.825	**<0.001**	0.891
	MgB	0.38	0.33	0.33	0.36	0.33	0.75	0.51	0.45	0.54	0.52				
**NEFA** **(mmol/L)**	Control	0.24	0.28	0.33	0.39	0.80	0.73	0.55	0.43	0.28	0.17	0.075	0.355	**<0.001**	0.937
	MgB	0.18	0.23	0.28	0.30	0.77	0.66	0.44	0.40	0.22	0.14				
**Glucose** **(mmol/L)**	Control	3.91	3.82	3.93	3.77	6.28	2.98	3.19	3.39	3.39	3.76	0.194	0.586	**<0.001**	0.111
	MgB	3.94	3.97	3.94	3.92	5.81	3.07	3.22	3.34	3.40	3.59				
**Haptoglobin (g/L)**	Control	0.40	0.40	0.41	0.42	0.43	0.66 ^a^	0.52 ^a^	0.59 ^a^	0.60 ^a^	0.51 ^a^	0.025	**<0.001**	**<0.001**	**<0.001**
	MgB	0.38	0.39	0.38	0.41	0.42	0.50 ^b^	0.43 ^b^	0.43 ^b^	0.45 ^b^	0.39 ^b^				
**Ionized Ca (mmol/L)**	Control	1.21	1.20	1.20	1.21	1.08	1.14	1.15	1.15	1.15	1.10	0.01	0.922	**<0.001**	0.943
	MgB	1.20	1.21	1.20	1.20	1.10	1.14	1.15	1.15	1.14	1.10				

^1^ Means with different superscripts (^a,b^) indicate significant difference between groups.

**Table 4 vetsci-10-00276-t004:** Prepartum and postpartum rumination time calculated as area under the curve (means ± SD) in dairy cows with (MgB, n = 34) and without (Control, n = 31) prepartum magnesium butyrate supplementation.

Rumination Time Parameters ^1^		Group		*p* Value ^2^
	Units	Control	MgB	
Baseline	min/d	351.5 ± 46.4	347.0 ± 45.5	0.850
AUC _PRE_	min	8582.6 ± 1210.4	9568.0 ± 1320.5	0.456
AUC _POST_	min	22 390.5 ± 3432.3	28 235.4 ± 4067.2	**0.012**

^1^ Baseline = the mean value of the times spent ruminating calculated for the first four days of the measurement; AUC _PRE_ = area under the curve calculated for the last 23 days before calving; AUC _POST_ = area under the curve calculated for the first 70 DIM. ^2^ Statistical significances are based on the Welch’s two-sample *t* test.

## Data Availability

The data presented in this study are available on request from the corresponding author.

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
