# Peer review of "Effect of Prepartum Magnesium Butyrate Supplementation on Rumination Time, Milk Yield and Composition, and Blood Parameters in Dairy Cows"

_vetsci, 2023, doi:10.3390/vetsci10040276_

Round 1

Reviewer 1 Report

This study investigated the effects of prepartum magnesium butyrate (MgB) supplementation on lactational performance, blood energy analytes, rumination time, and inflammation in dairy cows. Milk yield, milk composition, somatic cell counts, and rumination time were measured in MgB supplemented and unsupplemented cows. Blood samples were taken and analyzed for various biomarkers. Results showed that MgB supplementation increased milk yield, improved milk fat and protein concentrations, reduced somatic cell counts, and lowered haptoglobin levels. No significant differences were observed in blood biomarkers between groups. Postpartum rumination time was also improved in the MgB group. These findings suggest that prepartum MgB supplementation can improve lactational performance, reduce inflammation, and enhance postpartum rumination time in dairy cows. Overall, this study provides important insights into the potential benefits of MgB supplementation in dairy cows. Here are some specific comments that could be improved.

1.     Animals and Ethical Approval. The sentences describe the context and setting of the experiment, including the location, herd size, and enrollment criteria for the cows. However, more information on the study design and sample characteristics would be helpful for a more comprehensive evaluation.

2.     Although Diet Analysis section provides a clear description of the methods used to analyze the chemical composition of the prepartum and postpartum diets. However, it lacks important details such as the number of samples analyzed, the type of feed materials used, and the specific AOAC and Van Soest methods employed. It would also be helpful to include information on the accuracy and precision of the analytical methods used. Additionally, it would be beneficial to provide a brief explanation of why the various components of the feed were analyzed (e.g., to determine if there were differences in nutrient composition between the prepartum and postpartum diets).

Reviewer 2 Report

The experiment is of interest for the readership of Veterinary Sciences. Generally, the study was conducted in a clear and sufficient way. The three main drawbacks/issues are:

1)      No covariate period was included in the study/statistical model. From my experience, this is mandatory as cows have indeed individual variation that can affect the outcome. We saw this in several of our studies

2)      No feed intake data was collected, which is out outstanding importance and impedes the interpretation of the results in regards to the effect of MgB as well as it makes the rumination data only marginally important

3)      I think authors try to oversell the product. For instance, they always refer to “highly bioactive Mg” but do not reveal the exact Mg source or anything. This should be removed unless no further information on the actual bioavailability is given. Also, in L78 authors use the broad statement of butyrate is improving the immune system.

M&M:

·         Is there a justification for the dosage of 1.5% of DM?

·         How was the MgB included in the diet? Via premix or in concentrate pellets (btw, it remains unclear if concentrate was fed pelleted or not) or just as top-dressing?

·         Feeding of cows is not described, please include. How many times a day etc.

·         Please state the blood collection time in relation to feeding time. This will affect blood metabolites

·         Some details on blood processing, e.g. spinning speed of centrifuge, are also not stated

·         Did you detect any calving difficulties?

·         Statistics:

o   I not really agree with the model. The random effect is not acknowledging for repeated measurements as authors claim. Data that was collected repeatedly from the same animals should be considered by a repeated measurements statement appropriately

o   Did you run any outlier test?

o   Please indicate which post-hoc test was used

Results:

·         Table 3 is not complete, so I cannot fully assess it.

·         What is the benefit of AUC for rumination?

·         The milk composition data is very surprising. Why are milk fat levels in MgB cows so extremely high? Over 6%, was colostrum analysed? Because also milk protein is quite high. You may also discuss/consider that only morning milk was collected that is usually higher in fat..

·         The rumination data is also strange. 200-250 min rumination per day after 2-3 wks postpartum is so low and suggests rumen acidosis or other disorders. Again, DMI data would help a lot to actually understand if MgB cows ruminated more both BEFORE and AFTER calving due to a higher feed intake (when you relate rumination data to feed intake). This would explain why milk yield was higher in those cows. Still, these very low values for CON cows after calving make me wonder if the data collection worked ok.

Discussion:

·         Authors are often overselling their product in my opinion. Some more cautious wording is needed here as it is also in parts superficial

·         L350: How is the “highly bioavailable” increasing the milk yield?

·         L370-373: Is there a dose-dependent effect? As no feed intake was measured, authors do not know how much of the MgB was taken up. So it is a black box more or less

·         L382: iCa blood levels

·         L388-393: Lactose concentration is generally very stable as it has an osmolality impact + what extreme and unusual feeding situations do you mean?

·         L397: SSC are on a normal level for all cows. This must be stated.

·         The Hp part is exaggerating. Authors only measured one APP and not as well SAA or specific markers, too, but talk about systemic inflammation in CON cows. All Hp levels were low in general and it may depend on birth complications etc. then you need to argue with this. Also sub-acute inflammatory processes are not supported by the data in my opinion. At least, word it more cautious and less speculative.

·         Less stress in cows to more rumination is not true here, I think. Again, as no feed intake data is present, one can only speculate. But I can imagine that MgB cows had higher DMI and so more rumination and finally more milk production.

·         L466: the lacking feed intake data is a central issue and should not be reported as a closing sentence of the discussion.

Conclusions:

·         Again, rephrase it more cautiously. There is no indication of any sub-acute inflammatory processes

·          

Round 2

Reviewer 2 Report

I thank the reviewers for carefully addressing my comments and thoughts. The manuscript can be published in its present form, in my opinion.

Fyi regarding Table 3: it was cut in the original submission I received and therefore I could not read it. Now it is in landscape in the R1 version and it is absolutely fine.